# Neuroinflammation Modulation via α7 Nicotinic Acetylcholine Receptor and Its Chaperone, RIC-3

**DOI:** 10.3390/molecules26206139

**Published:** 2021-10-11

**Authors:** Tehila Mizrachi, Adi Vaknin-Dembinsky, Talma Brenner, Millet Treinin

**Affiliations:** 1The Agnes Ginges Center for Human Neurogenetics, Department of Neurology, Hadassah University Hospital, Hebrew University Medical School, Jerusalem 91120, Israel; tehilati@gmail.com (T.M.); adembinsky@gmail.com (A.V.-D.); 2Department of Medical Neurobiology, Hadassah Medical School, Hebrew University, Jerusalem 91120, Israel

**Keywords:** cholinergic anti-inflammatory pathway, neuroinflammation, α7 nicotinic acetylcholine receptor, RIC-3, multiple sclerosis, experimental autoimmune encephalomyelitis, Parkinson’s disease, Alzheimer’s disease

## Abstract

Nicotinic acetylcholine receptors (nAChRs) are widely expressed in or on various cell types and have diverse functions. In immune cells nAChRs regulate proliferation, differentiation and cytokine release. Specifically, activation of the α7 nAChR reduces inflammation as part of the cholinergic anti-inflammatory pathway. Here we review numerous effects of α7 nAChR activation on immune cell function and differentiation. Further, we also describe evidence implicating this receptor and its chaperone RIC-3 in diseases of the central nervous system and in neuroinflammation, focusing on multiple sclerosis (MS) and its animal model, experimental autoimmune encephalomyelitis (EAE). Deregulated neuroinflammation due to dysfunction of α7 nAChR provides one explanation for involvement of this receptor and of RIC-3 in neurodegenerative diseases. In this review, we also provide evidence implicating α7 nAChRs and RIC-3 in neurodegenerative diseases such as Alzheimer’s disease (AD) and Parkinson’s disease (PD) involving neuroinflammation. Besides, we will describe the therapeutic implications of activating the cholinergic anti-inflammatory pathway for diseases involving neuroinflammation.

## 1. Introduction

Nicotinic acetylcholine receptors (nAChRs) belong to a large and diverse family of acetylcholine-gated cationic ion channels. In mammals this family is composed of nine alpha subunits, whose N-terminus contains two disulfide linked vicinal cysteines enabling ligand binding, and seven non-alpha subunits. Most nAChR subunits assemble to form heteromeric receptors whereas α7 and α9 nAChR subunits can form homomeric receptors. Folding assembly and trafficking of mature nAChRs from the endoplasmic reticulum to the plasma membrane is a complex process which requires assistance from multiple cellular factors. Several proteins were shown to affect functional expression and properties of nAChRs including Resistance to Inhibitors of Cholinesterase-3 (RIC-3) and NACHO [1,2].

The nAChRs express widely and have multiple functions. In skeletal muscle nAChRs mediate excitation in the neuromuscular junction. Similarly, in the autonomic nervous system nAChRs are needed for excitatory synaptic transmission [3,4]. Functions of nAChRs, however, are not limited to synapses and to excitatory synaptic transmission. In the central nervous system (CNS) nAChRs have mostly modulatory roles, including a role in regulating release of neurotransmitters [5]. Moreover, these receptors are expressed in many non-excitable cells where they have been shown to affect migration, differentiation and proliferation [6].

CNS-expressed nAChRs have been implicated in several neurodegenerative diseases. Evidence supporting involvement of nAChRs in these diseases includes epidemiological studies linking some of these diseases to tobacco smoking and gene expression studies. Specifically, smoking reduces the prevalence of Parkinson’s disease (PD) [7]; schizophrenia is associated with heavy smoking [8]; and decreased brain expression of nAChRs was shown in Alzheimer’s disease (AD), PD, Lewy-body dementia, and schizophrenia [9,10,11,12], but the mechanisms underlying involvement of nAChRs in these diseases are, yet, unclear.

This review will focus on the Immunomodulatory roles of nAChRs. Neuronal nAChR subunits (subunits that are expressed in neurons and are not part of the skeletal muscle nAChR) are expressed in immune cells and in glial cells (microglia and astrocytes). Specifically, expression analysis in mice showed that microglia express all nine neuronal nAChR subunits examined while other immune cells express subsets of these subunits. Two of these subunits α9 and β2 showed high expression in all of the cells examined except regulatory T-cells (Treg). α5, α6, α7 and β4 subunits showed significant, although weaker, expression in several cell types (CD4^+^, CD8^+^, CD11c^+^ and CD11b^+^), while the expression of α3 and β3 was limited to microglia [13].

In immune cells, nAChRs have immunomodulatory roles and among them the best studied role is of the α7 nAChR as part of the cholinergic anti-inflammatory pathway. This pathway was first described in the experimental model of septic shock. In this model, inflammatory activation is induced by lipopolysaccharides (LPS) injection leading to increased production of pro-inflammatory cytokines (such as IL-1 and TNF-α), micro-circulatory failure and systemic hypotension. Vagal stimulation attenuated the pathology associated with septic shock an effect requiring α7 nAChR, whose activation in macrophages inhibited pro-inflammatory cytokine production from these cells. Importantly, α7 nAChR was required for the inflammation reducing effects of vagal stimulation (Figure 1A left and [14,15]). More details on the cholinergic anti-inflammatory pathway are provided below.

Additional immunomodulatory effects of activating immune cell-expressed nAChRs are emerging. One such pathway depends on α9 and α10, with α7 having a minor role. This pathway inhibits the pro-inflammatory effects of ATP on monocytes thus reducing IL-1β release from these cells [16]. Interestingly, C-reactive protein (CRP)–a blood marker for inflammation that is produced by the liver in response to IL-1β–is an agonist for this nAChR [4]. These findings suggest a nAChR-dependent negative feedback loop reducing IL-1β release and the deleterious effects resulting from its overproduction (Figure 1B).

Recent findings identified a pro-inflammatory pathway involving nAChRs and acetylcholine. Activity of this pathway is initiated by neurons in the central nucleus of the amygdala (CeA) and in the paraventricular nucleus (PVN) of the hypothalamus. Both types of neurons express the stress related corticotropin-releasing hormone (CRH). Activation of these neurons via stimulation of the sympathetic splenic nerve leads to acetylcholine (ACh) release from T-cells which activates B-cell expressed α9 nAChR to promote plasma cell formation and IgG antibody production (Figure 1A right and [17]).

## 2. α7. nAChR and the Cholinergic Anti-Inflammatory Pathway

α7 nAChR, a key player in the cholinergic anti-inflammatory pathway, was first identified as the α-bungarotoxin (α-BTX) binding receptor in the CNS [18]. This receptor has high permeability to calcium, likely to enable some of its effects on cellular functions [19]. In addition, α7 receptors are activated by ACh and by choline, a precursor and breakdown product of ACh as well as a breakdown product of membrane phospholipids. This enables α7-mediated responses to both targeted transmission and to localized tissue damage.

In immune cells α7 nAChR activation leads to phosphorylation and activation of Janus kinase 2 (JAK2) and to phosphorylation and nuclear entry of Signal Transducer and Activator of Transcription 3 (STAT3) which inhibits expression of pro-inflammatory cytokines. In parallel α7 nAChR activation reduces degradation of IkBα therefore inhibiting nuclear translocation of NF-kβ and the expression of pro-inflammatory cytokines. Interestingly, multiple lines of evidence suggest metabotropic activity of α7 nAChR in immune cells. Mechanisms enabling the immune modulating effects of α7 nAChR are reviewed in [20].

## 3. Cholinergic Signaling via nAChRs in Immune Cells

The cholinergic anti-inflammatory pathway was initially described by its ability to inhibit inflammation through activation of macrophage expressed α7 nAChR [14,15,21]. Below we provide detailed description on the immunomodulatory effects of activating nAChRs in different immune cell types, including CNS resident glial cells having immune functions. A summary of this information is provided in Figure 2.

### 3.1. α7. nAChR in the Periphery

#### 3.1.1. Macrophages

The α7 nicotinic receptor expressed in macrophages is involved in the regulation of inflammatory processes. Activation of α7 nAChR using acetylcholine, nicotine or other agonists decreases pro-inflammatory cytokine production in response to LPS. Unlike production of anti-inflammatory cytokines which is unaffected by these agonists. The mechanism implicated was reduced NF-κB-mediated transcription. [14,22] Involvement of NF-κB-mediated transcription in this pathway is consistent with its anti-inflammatory effects, as many pro-inflammatory mediators are transcribed by this transcription factor [23].

In addition to suppressing the inflammatory response to LPS, activation of α7 nAChR suppressed pro-inflammatory cytokine production in response to activation of various toll-like receptors (TLRs) such as TLR2, TLR3, and TLR9 [24].

The anti-inflammatory effects of ACh are mimicked by nicotine, but not by muscarine, an agonist of muscarinic acetylcholine receptors, and these effects depend on α7 nAChR. These effects are eliminated when macrophages were exposed to antisense oligonucleotides specific for the α7 nAChR or in loss of function mutants for the gene that codes for this receptor (*CHRNA7*) [15,22].

Further evidence for anti-inflammatory effects of activating macrophage expressed α7 nAChRs were provided in experiments using a specific activator of this receptor, the allosteric agonist and positive allosteric modulator (ago-PAM) GAT107. Application of this drug following LPS stimulation of a murine macrophages cell line (RAW264.7) significantly reduced IL-6 secretion by these cells. These anti-inflammatory effects were eliminated by methyllycaconitine (MLA), a specific antagonist of α7 nAChR [25].

#### 3.1.2. Dendritic Cells

Mashimo et al. (2020) [26] established the role of the cholinergic inflammatory system in dendritic cells (DCs), reporting that these cells express nAChR subunits, including α7 nAChR. DCs can produce ACh and express the enzymes for ACh synthesis and degradation, choline acetyltransferase (ChAT) and acetylcholine esterase (AChE). In addition, they express Secreted Ly-6/uPAR-related protein 1 (SLURP-1), the endogenous peptide that acts as a positive allosteric ligand for α7 nAChR. Overall, DCs express most of the components necessary for a functional cholinergic system [26,27].

We and others have shown decreased antigen presentation following cholinergic activation in vivo [28,29]. Additionally, following nicotine treatment, the expression of co-stimulatory molecules such as CD80, CD86 and MHCII on antigen presenting cells (APCs) was reduced [30]. In vitro exposure of human DCs to nicotine decreased secretion of pro-inflammatory cytokines and reduced DCs phagocytic activity. Moreover, a direct effect of nicotine on DCs was also found when DCs were cultured with nicotine and failed to mount an effective Th_1_ response in T-cells [31].

#### 3.1.3. T-Cells

Proliferation of specific T-Cells is crucial to mount an effective adaptive immune response. Murine and human T-cells express α7 nAChR, and this expression is increased following antigenic or mitogenic stimulation [28,32]. Our recent study on Experimental Autoimmune Encephalomyelitis (EAE) mice treated with the α7 nAChR ago-PAM, GAT107 showed reduced ex-vivo antigen specific induced T-cell proliferation and reduced secretion of pro-inflammatory cytokines (IL-17, IL-6 and IFNγ) while secretion of the anti-inflammatory cytokine, IL-10, was increased [25]. A similar result of reduced T-cell activity was shown when cholinergic agents such as Rivastigmine, an acetylcholine esterase inhibitor (ACHEI), or EN101 (an anti-sense oligonucleotide to exon 2 of AChE) were used. Blocking of α7 nAChR by using pharmacology and anti-sense oligonucleotides eliminated this effect [28,33]. Exposure to AChEI increases extracellular ACh concentration and induces cholinergic receptor activation. This mechanism is similar to the therapeutic mechanism by which AChEIs function in myasthenia gravis (MG) or Alzheimer’s disease (AD).

A recent study using an experimental inflammatory bowel disease model described nicotine-induced suppression of the Th_2_ lineage activity and an increase in Th_1_ and Th_17_ activity [34]. Notably, this effect of nicotine was correlated with the expression level of the α7 nAChR: the inhibitory effect of nicotine on Th_2_ lineage was correlated with higher expression of α7 nAChR on these T-cells, while the failure to suppress the Th_1_ lineage correlated with lower expression of α7 nAChR on these cells. Thus, it seems that nicotine does not suppress a specific Th lineage, but rather may inhibit any active lineages as long as α7 nAChR is expressed on these cells.

Mashimo et al. demonstrated that α7 nAChRs expressed on T-cells and APCs have different effects on naïve CD4+ T-cell differentiation. Initially, activation of α7 nAChRs on APCs reduced antigen processing and thereby resulted in suppression of T-cell differentiation, but activation of α7 nAChRs on T-cells increased T-cell differentiation [35]. Immune activation of both naïve α7 nAChR knockout (KO) and of wild-type CD4+ T-cells via TCR/CD3 activation increased their differentiation into T regulatory cells (Tregs) and effector T-cells. Furthermore, the α7 nAChR agonist, GTS-21, further enhanced the differentiation of wild-type naïve CD4+ T-cells into Tregs and effector T-cells, but without any effect on cells that do not express the α7 nAChR. These results confirm that these effects on T-cell differentiation are mediated by α7 nAChR. T-cells and APCs (DCs and macrophages) express ChAT and therefore synthesize ACh, and their immunological activation enhances ACh synthesis [26,36]. ACh released from these activated immune cells acts via α7 nAChRs in an autocrine and/or paracrine manner. Thus, α7 nAChRs on T-cells and APCs is likely to be part of an auto-inhibitory mechanism [37].

Tregs from naive mice express α7 nAChR, and its activation by nicotine enhanced the inflammation suppressing capacity of these cells. Nicotine stimulation up-regulated the expression of cytotoxic T-lymphocyte-associated antigen (CTLA)-4 and forkhead/winged helix transcription factor p3 (Foxp3) on Tregs [38]. Our recent study also shows that activation of α7 nAChR up-regulated Foxp3 expression and, therefore, supports the role of α7 nAChR in regulating the immunomodulatory effects of Tregs during neuroinflammation [25].

#### 3.1.4. B-Cells

Several studies have showed that B-cells express nAChR subunits [39,40,41]. Mice deficient for both α7 or β2 nAChR subunits had less IgG-producing cells, and less serum IgG. However, cells from deficient mice produce more antibodies in response to activation, caused by increased CD40 expression. Interestingly, the inhibitory effects of nicotine on antibody production could only be demonstrated in the absence of the β2 subunit [42]. In another study, α7 deficient mice had increased serum IgG1 levels, increased specific IgG1 production in response to antigen, and higher production of IL-6, TNF-α and IFN-γ by T-cells from the same mice. This increased antibody production can be attributed to the effect of α7 on CD4 T-helper cells, which induce B-cell antibody production [43]. Overall, there is evidence that different nAChR subtypes play non-identical roles in B-lymphocytes [44].

Recently, we reported that the α7 ago-PAM GAT107 also affects B-cell function [25]. Mice with EAE showed a significant reduction in B-cell markers and lower pathogenic antibody production, following GAT107 treatment. B-cell activating factor (BAFF) and A proliferation-inducing ligand (APRIL) as well as B220, were significantly reduced by the activation of α7 nAChR. BAFF is a potent survival factor for B-cells that plays an essential role in the preservation and maturation of peripheral B-cells and APRIL is important for B-cell development. In addition to expressing nAChRs B-cells express ChAT mRNA, AChE mRNA and contain ACh. Also, activation of TLR by LPS in murine B cells, led to activation of cholinergic function by enhancing ACh synthesis [26].

#### 3.1.5. Natural Killer Cells

Natural killer (NK) cells play a cytotoxic role against target cells and modulate the innate and the adaptive response via dendritic cells. Zanetti et al. (2016) showed that NK cells express the α7 nAChR, and that its expression was increased following activation by cytokines (IL-12, IL-18, and IL-15). It is important to note that the α7 nAChR is functional in NK cells, since activation with a α7 nAChR agonist increases intracellular calcium concentration, mainly released from intracellular stores. In addition, activation of α7 nAChR in Human NK Cells regulates NKG2D expression (a NK cell activating receptor).

NK cells that were exposed to specific α7 nAChR agonist-PNU-282987 display significantly reduced cytotoxic activity. Also, while NK cells are stimulated with cytokines (IL-12, IL-18, and IL-15) and exposed to α7 nAChR agonist- PNU-282987, the percentage of IFNγ producing NK cells was lower. Those effects of PNU-282987 were abolished in the presence of α-BTX, an antagonist of α7 nAChR [45].

### 3.2. α7 nAChR in CNS Glial Cells

#### 3.2.1. Microglia

Similar to the findings in macrophages, the same effects of nicotine can be demonstrated on microglia, the brain-resident macrophages [46]. Microglia express both α7 nAChR mRNA and protein. Furthermore, microglia in co-culture with neurons increased pro inflammatory factor secretion following LPS stimulation. And, ACh reduced secretion of pro inflammatory factors and reduced the pro-apoptotic effects of activated microglia on neurons. These effects of ACh were abolished when α7nAChR was silenced using α7 nAChR-shRNA [47].

LPS-stimulated microglia that were also exposed to nicotine or ACh showed reduced TNF-α release. This reduction was mediated through MAP-kinase inhibition. Of note, reduction in TNF-α release by microglia was demonstrated for α7 nAChR selective ligands that are poor channel activators or even antagonists of channel activity; these effects suggest that in microglia α7 nAChR does not function as an ion channel [48].

Neuroinflammation induced by LPS in primary fetal sheep microglia cultures resulted in pro-inflammatory microglial phenotype. Exposure to an agonist of α7 nAChR reversed this phenotype. Conversely, an antagonist of α7 nAChR potentiated the pro-inflammatory microglial phenotype. Hepcidin-ferroprotein signaling and iron homeostasis were implicated in maintaining the microglial inflammatory phenotype and in the effects of α7 nAChR signaling on this phenotype [49].

Over activation of microglia is associated with aggravation of EAE in mice [50]. Ke et al., showed that activating α7 nAChR can inhibit NLRP3 inflammasome via the regulation of β-arrestin-1 in microglia, contributing to the suppression of neuroinflammation and attenuation of EAE severity [51]. Similarly, we have shown that treatment of EAE mice with the ACEI, rivastigmine, led to clinical improvement that was associated with reduced activation of microglia and preservation of neurofilaments [28].

#### 3.2.2. Astrocytes

The α7 nAChRs are also expressed by astrocytes [52] and there is evidence that those receptors can affect synaptic transmission in the CNS [53]. As activation of α7 nAChR increased expression of post synaptic α-amino-3-hydroxy-5-methyl-4-isoxazolepropionic acid (AMPA) receptors in hippocampal GABAergic neurons. This led to increased AMPA receptor mediated synaptic currents which in turn can reinforce the activity of specific GABAergic synapses. In addition to these neuromodulatory effects, astrocyte expressed α7 nAChR has immunoregulatory roles; using a α7 nAChR agonist, GTS21, in astrocyte cultures stimulated with LPS, attenuated IL-6 and TNF-α secretion, as well as reduced inflammatory astrocyte induced apoptosis of neuronal cells. This effect was reversed by the α7 nAChR antagonist MLA, indicating the specificity of this response [52].

Further support for the neuroprotective effect of α7 nAChR in astrocytes is provided by the α7 nAChR agonist PNU-282987, which reduced apoptosis of neurons in glia-neuron co-culture and reduced glutamate induced toxicity of these neurons. These effects were achieved via inhibition of TGFβ signaling. In addition, activation of the α7 nAChR on astrocytes resulted in decreased neurotoxicity of the chemotherapeutic agent (oxaliplatin) [54].

Cao et al. performed RNA-Seq analysis in primary fetal astrocyte cultures exposed to LPS together with a selective α7 nAChR agonist or antagonist. Stimulation by a α7 nAChR agonist promoted the neuroprotective profile of astrocytes, and α7 nAChR inhibition reduced it; LPS double-hit (first in vivo, then in vitro) on astrocytes exacerbates these effects [55].

## 4. α7 nAChR and CNS Diseases

In mice loss of function mutations in subunits contributing to major CNS expressed nAChRs (α7, α4 and β2) have no easily discernible phenotypes [56]. But, in humans copy number variations of the gene encoding for α7 nAChR subunit (*CHRNA7*) are associated with brain diseases such as epilepsy or autism [57]. In addition, a 2bp deletion in of the gene encoding for a human-specific chimeric protein, CHRFAM7A, containing a duplicate of exons 5-10 of *CHRNA7*, shows strong linkage to schizophrenia [58]. Instability of the genomic region harboring the *CHRNA7* complicates its analysis, as disease causing deletions encompassing this gene also cover additional genes. Since, heterozygosity for rare small deletions covering *CHRNA7* alone is associated with similar phenotypes to those associated with larger deletions in this region it is likely that haplo-insufficiency of this gene is a cause for disease in humans [57], but some heterozygous carriers of *CHRNA7* containing deletions are phenotypically normal. This confounding finding may be explained by incomplete penetrance of these deletions due to interactions with the genetic background or with the environment [57].

The CHRFAM7A chimeric protein, which is unique to humans, was shown to interact with and inhibit surface expression of α7 nAChRs [58,59]. This chimeric protein expresses in immune cells and its expression in these cells is regulated by LPS, nicotine and IL1β. Thus, this protein is likely to be a human specific regulator of the cholinergic anti-inflammatory pathway [58,59].

Of note, involvement of α7 nAChR in neurodegenerative diseases may stem, at least partly, from its immunomodulatory effects, which in the CNS are likely to involve astrocytes and microglia [60]. As described above, α7 nAChR is expressed in glial cells and was shown to affect their activity.

## 5. RIC-3 and α7 nAChRs in Neurodegenerative Diseases

RIC-3 was first characterized in *C. elegans* as an ER-resident protein that is needed for the maturation of multiple nAChRs but not for the maturation other ligand-gated ion channels [1]. Later analysis showed conservation of RIC-3 sequence and function during evolution [61]. Moreover, RIC-3 is required for heterologous expression of the α7 nAChR in non-neuronal mammalian cells [62].

Analysis of *ric-3* loss-of-function mutants in *C. elegans* demonstrated reduced functional expression of multiple nAChRs in these mutants. Therefore, RIC-3 was suggested to promote surface expression of co-expressed nAChRs [1], but later analysis demonstrated both positive and negative effects of RIC-3 on co-expressed receptors depending on identity of the co-expressed receptor and on the expression system [61,63]. One explanation for these different effects is the finding that altering RIC-3-to-nAChR ratio shifts its effects from positive to negative [64,65].

RIC-3 was expected to promote expression of α7 nAChR in-vivo, as functional expression of α7 nAChR in non-neuronal cells requires co-expression with RIC-3 [62] and expression patterns of *ric3* and the α7 nAChR subunit overlap [61]. However, knockout of *ric3* did not significantly affect α7 nAChR expression in the brains of mice [66]. Nevertheless, its knockdown using siRNA-mediated silencing eliminated the anti-inflammatory effects of GAT107, an α7 nAChR-specific agonist [67]. This suggests cell-specific requirement of RIC-3 for functional expression of α7 nAChR.

RIC-3 like α7 nAChR has been implicated in diseases of the central nervous system: *ric3* expression was increased in bipolar disease and in Schizophrenia [68]. Genome wide association studies (GWAS) identified polymorphisms in the *ric3* region in association with multiple sclerosis (MS) [69,70]. GWAS also suggested that ric3 variants modify cognitive maintenance in the elderly [71]. Mutations in *ric3* were found in PD patients [72].

## 6. α7 nAChR and RIC-3 in Neurodegenerative Diseases Involving Neuroinflammation

The well-established role of α7 nAChR in the cholinergic anti-inflammatory pathway suggests that α7 may be involved in neuroinflammatory diseases, as reviewed by [20,60]. RIC-3, like α7 nAChR is expressed in immune cells. Expression of *ric3* is regulated by pro-inflammatory stimuli, and its knockdown eliminates the anti-inflammatory effects of a α7 nAChR-specific agonist [65,67]. Thus, RIC-3 like α7 nAChR is likely to be involved in neuroinflammatory diseases. Below we will provide evidence implicating the cholinergic anti-inflammatory pathway, α7 nAChR and RIC-3 in three neurodegenerative diseases: MS, AD and PD. While neuroinflammation is causal for MS its role in AD and PD is less clear. Nevertheless, neuroinflammation is likely to contribute to progression of these diseases [73]. Therefore, we will also describe attempts to harness the cholinergic anti-inflammatory as therapy for these diseases.

## 7. Involvement of α7 nAChR and RIC-3 in EAE

EAE is the animal model used to study MS, and for the study of neuroinflammation. Using the chronic EAE model in C57bl mice we showed that treatment of EAE with EN101, an anti-sense oligodeoxynucleotide, targeted to AChE mRNA, reduced EAE clinical severity and CNS inflammation intensity. This clinical amelioration was accompanied by reduction in lymphocyte proliferation and reduced inflammatory infiltrates to the spinal cord. These results suggest that AChEIs increase the concentration of extracellular ACh, rendering it available for interaction with a nicotinic receptor expressed on lymphocytes [32].

In addition, a marked improvement in EAE severity was found by treatment with rivastigmine. Rivastigmine is a pseudo-irreversible AChEI that was shown to ameliorate cognitive dysfunction in AD [74]. Rivastigmine treatment of EAE mice ameliorated disease severity, reduced demyelination, and showed a multi-level activity on various stages of the immune response in EAE, as well as improving cognitive function [28].

In another set of experiments, we found that treatment with bifunctional compounds comprising both a cholinergic up-regulation moiety (the AChEI pyridostigmine or cytisine) and a nonsteroidal anti-inflammatory moiety (ibuprofen), ameliorated clinical symptoms in the same EAE model [32,75]. In this setting the presence of the cholinergic up-regulation moiety increased the anti-inflammatory effects and reduced the pro-inflammatory effects.

Nicotine administration also ameliorated EAE severity and reduced T-cell proliferation in response to encephalitogenic antigens, as well as the production of cytokines by Th1 cells (TNFα and IFNγ) and Th17 cells (IL-17, IL-17F, IL-21, and IL-22) [29].

Recently, we examined effects of GAT107, a α7 nAChR specific ago-PAM, on EAE mice. Administration of GAT107 to EAE mice, resulted in EAE amelioration, accompanied by reduced inflammatory infiltrates to the spinal cord. In addition, the expression of immune cell markers was altered by GAT107 treatment, which induced a significant reduction in macrophages, dendritic cells, and B cells, as well as a reduction in the production of specific antibodies [25].

In addition to our results, others have reported that treatment with nicotine ameliorates EAE severity, reduced CNS infiltrates, and reduced T-cell reactivity in this model [13,76]. Of note, some effects of nicotine on EAE were attributed to α9 and β2 containing nAChRs. These receptors were suggested to play different roles from α7 nAChR in endogenous ACh- and in nicotine-dependent modulation of immune functions [77]. Table 1 summarizes information from studies using the EAE model to examine the therapeutic effects of activating the cholinergic anti-inflammatory pathway.

Besides, therapeutic effects of cholinergic agents on EAE, we showed that induction of EAE in mice is associated with dynamic changes in expression of *ric3*, encoding for RIC-3, in the periphery and in the CNS. Furthermore, similarities in the expression dynamics of *ric3* and *CHRNA7*, encoding for the α7 nAChR subunit, were found. Moreover, *ric3* was found to be required for the anti-inflammatory effects of cholinergic agonists in RAW264.7 cells. [67].

## 8. α7 nAChR, RIC-3 and Multiple Sclerosis

The beneficial, clinical, immunological and neuropathology reducing effects of activating the cholinergic anti-inflammatory pathway in EAE mice, led to several studies examining this pathway in MS patients. This analysis showed increased responsiveness of peripheral blood mononuclear cells (PBMCs) to the anti-inflammatory effects of nicotine, as shown by its effects on PHA-dependent pro-inflammatory cytokine release. This enhanced responsiveness correlated with increased expression of α7 nAChR in PBMCs stimulated with nicotine and PHA in MS patients but not in healthy donors [78] Moreover, treatment of MS patients with IFN-β led to reduced serum levels of pro-inflammatory cytokines and increased serum ACh levels; suggesting that circulating cytokines and ACh are co-regulated [79]. In addition, levels of ACh and of the enzymes responsible for its synthesis and degradation were altered in the serum of MS patients [79,80]. Analysis of NK cells from MS patients showed increased intracellular ACh levels. This increase correlated with disease severity as seen by the EDSS score, lesion number and lesion volume [81]. Together, these changes suggest compensatory mechanisms occurring in MS patients aimed at reducing inflammation via enhanced activity of the cholinergic anti-inflammatory pathway. Previous reports showed low levels of ACh in the CSF and in the serum of MS patients when compared to healthy controls; this effect is opposite to the high levels of pro-inflammatory cytokines detected in these patients, supporting a relationship between pro-inflammatory cytokines and ACh levels in MS patients [82]. 

The beneficial effects of nicotine on disease pathology in the animal model for MS suggested therapeutic potential for nicotine in MS patients. Tobacco smoking, however, is a well-known risk factor for MS [83]. Although, an epidemiological study undertaken in Sweden showed reduced MS risk in users of moist snuff, also a source of nicotine [84]. Immunomodulatory effects of tobacco smoke-derived nicotine are implicated as a risk factor for MS by a study showing that polymorphisms in loci encoding for α7 and α9 nAChRs modify the association between smoking and MS [85]. Alternatively, it was suggested that while tobacco-derived nicotine ameliorates MS pathology other components of tobacco smoke have detrimental effects. Evidence supporting this suggestion were provided from analysis of the EAE model for MS [86].

RIC-3 the chaperone for α7 nAChR was also implicated in MS. Genome-wide association studies detected an association between single nucleotide polymorphisms in the region encoding for this protein and MS or specific types of MS lesions [69,70]. The known role of the RIC-3 protein in α7 nAChR’s maturation, combined with the known roles of α7 nAChR in inflammation and its likely involvement in MS, suggest that genetic variants in the RIC-3 encoding gene (*ric3*) may affect MS via their effects on α7 nAChR’s maturation in immune cells. Support for this hypothesis comes from our recent experiments showing that siRNA-mediated knockdown of *ric3* in a macrophage cell line eliminated the anti-inflammatory effects of GAT107, a α7 nAChR specific agonist. In addition, in support of variable expression and function of *ric-3* and *CHRNA7* in some MS patients’, it should be noted that *ric-3* expression is increased in PBMCs from some MS patients and this increase correlates with increased expression of *CHRNA7* (encoding for α7 nAChR) [67]. Besides, GAT107 reduced pro-inflammatory cytokine (IL-6 and IL-17) secretion from PBMCs derived from MS and HD, and this cholinergic effect was more variable in the MS patients [25]. Future, work should examine how this increased expression of *ric3* and CHRNA7 affects MS progression and the relationship between the *ric3* polymorphisms associated with MS and *ric3* expression.

Currently treatment of MS at its neurodegenerative stage needs improvement. Findings showing impairment of the cholinergic anti-inflammatory pathway in MS and studies showing that activation of this pathway ameliorates inflammatory and neurodegenerative components of this disease (Table 1) support the cholinergic anti-inflammatory pathway as target for new MS drugs.

## 9. Parkinson’s Disease, Neuroinflammation and the α7 nAChR

Parkinson’s disease is a neurodegenerative disease whose hallmark is degeneration of dopaminergic neurons in the substantia nigra. A1 astrocytes which are a marker for neuroinflammation are detected in the substantia nigra of PD patients [73]. Besides, blocking microglia-dependent astrocyte activation into A1 astrocytes is neuroprotective in an animal model for PD [87]. Therefore, one explanation for the reduced prevalence of PD among tobacco smokers [7] is reduced neuroinflammation due to activation of the cholinergic anti-inflammatory pathway. Specifically, tobacco-derived nicotine was suggested to protect substantia nigra dopaminergic neurons via activation of microglia- or astrocyte-expressed α7 nAChRs leading to reduced neuroinflammation [88].

In support for the above explanation, nicotine protects substantia nigra dopaminergic neurons in animal models for PD in which dopaminergic neurons are selectively targeted using the neurotoxins 6-OHDA or MPTP. However, reduction of PD symptoms, can also be explained by effects of nicotine on nigro-striatal dopamine release which is mediated primarily by α6β2-containing receptors expressed in dopaminergic neurons; expression of this receptor is significantly reduced in animal models for PD [89].

α7 nAChR activation ameliorates both the degeneration of dopaminergic neurons and the associated neuroinflammation. For example, in mice nicotine reduced the motor deficits induced by MPTP and these effects were associated with reduced dopaminergic neuron degeneration. In addition, nicotine reduced astrocyte and microglial activation in the substantia nigra. Effects of nicotine on dopaminergic neuron degeneration and on inflammatory markers were abolished by an α7 nAChR antagonist [90]. Also, α7 nAChR specific agonists reduced degeneration of dopaminergic neurons and neuroinflammation in the striatum of rats treated with 6-OHDA [91] and in the subtantia nigra of mice treated with MPTP [92]. In addition, nicotine pretreatment decreased microglial activation and significantly reduced LPS-induced TNF-α release. Also, co-cultures of microglia and mesencephalic neurons, pretreated with nicotine had significantly decreased loss of tyrosine hydroxylase-immunopositive (TH-ip) cells (tyrosine hydroxylase is a marker for dopaminergic neurons). This effect was blocked by α-bungarotoxin, an α7 nAChR selective blocker. Besides, chronic nicotine pretreatment in rats had the same protective effect in the substantia nigra, suggesting that nicotine protects dopaminergic neurons via an anti-inflammatory mechanism [93].

## 10. Alzheimer’s Disease Neuroinflammation and the α7 nAChR

Alzheimer’s disease is a neurodegenerative disease and the most common cause of dementia in the elderly. Neurodegeneration in AD patients is associated with neuroinflammation. Specifically, reactive microglia and astrocytes migrate to and surround amyloid beta (Aβ) deposits, a hallmark of AD pathology. The roles of astrocytes, microglia and of neuroinflammation in AD progression is complex. While, microglia are responsible for clearance of toxic Aβ deposits and thus have neuroprotective roles, reactive microglia and astrocytes are likely to exacerbate pathology. Accordingly, various anti-inflammatory drugs have been tested for their therapeutic effects on AD, as reviewed in [94].

An early pathology associated with AD is reduced cholinergic signaling. This reduction led to the “cholinergic hypothesis of AD” and to the development of AChEI drugs as a therapy for AD. One explanation for the role of cholinergic reduction in AD pathology is reduced activity of the cholinergic anti-inflammatory pathway in these patients. This hypothesis suggests that treatment with AChEIs improves AD symptoms via reduced neuroinflammation. In support, AD patients treated with AChEIs show altered immune profile which is consistent with inflammation reducing effects of these drugs [95,96,97,98].

One AChEI used to treat AD, galantamine, is a relatively weak AChEI also having allosteric potentiating activity on nAChRs. This potentiation of nAChR activity may contribute to galantamine’s therapeutic effects. Indeed, galantamine increased Aβ phagocytosis by microglia, an effect depending on α7 nAChR and on the allosteric binding site for galantamine present in this receptor. Interestingly, in vitro depletion of choline, the degradation product of ACh and an agonist of α7 nAChR, abolished the Aβ phagocytosis inducing effects of galantamine. This result suggests that effects of galantamine on AD can be, at least partially, explained by potentiation of choline dependent enhancement of Aβ clearance by microglia [99].

Recently the therapeutic effects of activating the cholinergic anti-inflammatory pathway on AD pathology were examined in-vivo, in an APP/PS1 mouse model for AD [100]. This research showed that the positive allosteric modulator, JWX-A0108, improved spatial memory, reduced neurodegeneration and reduced neuroinflammation in this AD model. These effects were eliminated by a α7 nAChR specific antagonists, MLA. Together these results support activation of the cholinergic anti-inflammatory pathway as a therapy for AD. In support of the involvement of α7 nAChR in AD, treatment with an α7 nAChR specific antibody that reduces its expression induced inflammation, Aβ42 accumulation and impaired memory in the brains of mice [101].

Aβ was shown to interact with and activate α7 nAChR leading to investigations into the role of this receptor in AD, reviewed in [102]. Studies examining effects of α7 nAChR knockout on pathology and behavior of AD models led to conflicting results [103,104,105]. One recent paper suggested that oligomeric Aβ_42_ activated α7β2 nAChRs to enhance excitability of basal forebrain cholinergic neurons, a possible explanation for the network hyperexcitability and cognitive decline seen in AD patients [106]. Another study showed increased age dependent AD-like pathology and increased Aβ levels in α7 nAChR knockout mice; this suggests that α7 nAChR malfunction precedes Aβ pathology [105]. This study also showed increased astrocyte activation in α7 nAChR knockout mice [105]. The role of the cholinergic anti-inflammatory pathway in effects of α7 nAChR knockout on AD-like pathology is, yet, unknown.

Contribution of the cholinergic anti-inflammatory pathway to AD pathology in humans is, yet, unclear. Evidence supporting such involvement include: The proportion of α7 nAChR-expressing astrocytes, but not of α4 nAChR-expressing astrocytes, was increased in the hippocampus and entorhinal cortex of AD patients [107]. Furthermore, α7 nAChR expressing microglia were shown to accumulate around Aβ plaques in AD brains [99]. These findings suggest that in AD brains the cholinergic anti-inflammatory pathway functions to reduce Aβ deposits. We note, however, that tobacco smoking is a risk factor for AD [108,109]. But, there is no evidence linking tobacco smoking associated AD risk to alterations in the cholinergic anti-inflammatory pathway.

## 11. Conclusions

This review highlights the importance of α7 nAChR and its chaperon RIC-3 in the regulation of immune responses as part of the cholinergic anti-inflammatory pathway.

We describe the immune functions of this pathway on multiple types of immune cells and on glial cells. We also describe evidence for involvement of the non-neuronal cholinergic receptors in regulation of inflammatory processes in the experimental model of MS in which α7 nAChR agonists and antagonists have been shown to modify disease progression. Thus, α7 nAChR is a promising target for drugs aimed at downregulating inflammatory processes leading to or exacerbating various diseases including neurodegenerative diseases. Since AChEIs, (which induce cholinergic upregulation and activation of α7 nAChRs), are already in clinical use for several neurological indications, such as myasthenia gravis and AD, these agents can also be used for immonomodulation of other neurodegenerative diseases involving neuroinflammation. Furthermore, the ago-PAM GAT107 is promising candidates for treatment in these conditions, as we describe in this review. The widespread effects of the cholinergic anti-inflammatory pathway on inflammation in the periphery and in the CNS suggest the therapeutic potential of harnessing this pathway in diseases involving neuroinflammation. Better understanding of the mechanism of action of α7 nAChR in immune cells and of proteins, such as RIC-3, affecting α7 nAChR’s activity may facilitate development of treatments for neurodegenerative and other diseases involving inflammatory processes.

## Figures and Tables

**Figure 1 molecules-26-06139-f001:**
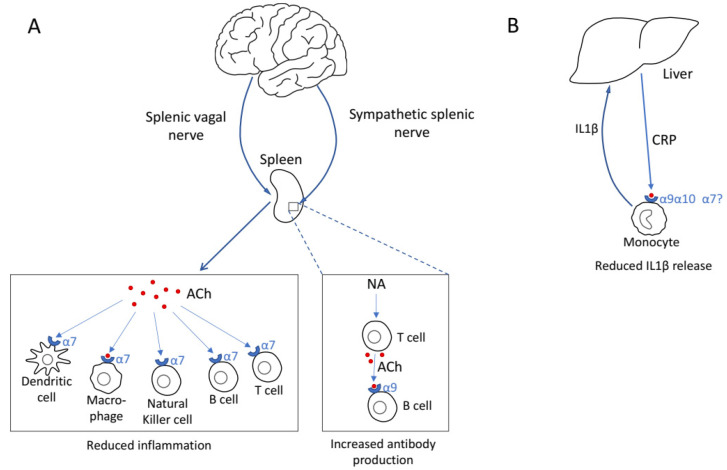
Immunomodulation mediated by nAChRs. (**A**) Left pathway, the splenic vagal nerve activity via the cholinergic anti-inflammatory pathways reduces inflammation within and outside the spleen. Right pathway, the sympathetic splenic nerve via neuroadrenaline (NA) release in the spleen promotes antibody production. (**B**) An anti-inflammatory negative feedback loop involving monocyte expressed nAChRs.

**Figure 2 molecules-26-06139-f002:**
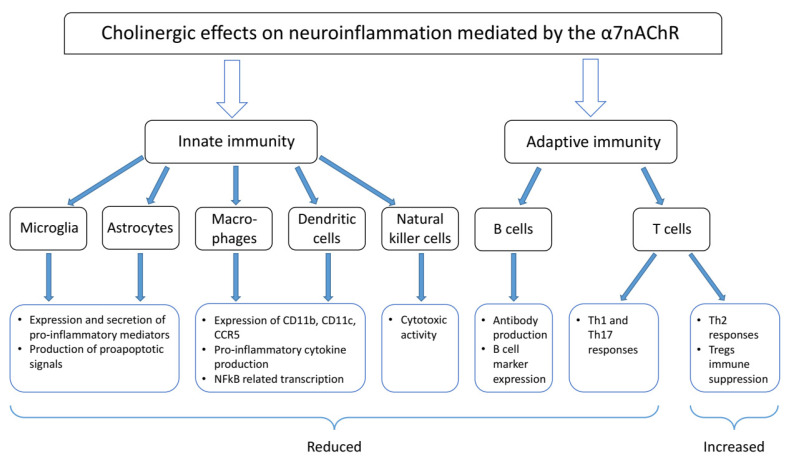
Cholinergic effects of α7 nAChR on immune and glial cells.

**Table 1 molecules-26-06139-t001:** EAE amelioration via α7 nAChR activation. Summary of studies using the EAE model to examine the therapeutic effects of activating the cholinergic anti-inflammatory pathway.

Cholinergic Agent Used for EAE Treatment	CNS Pathology	T-Cell Activity	B Cells	Reference
EN1O1	Reduced inflammatory Infiltrates in the spinal cord	Reduced proliferation and proinflammatory cytokine secretion	----	Nizri et al., 2005
IBU-PO IBU-PD	Reduced inflammatory activation of astrocytes in vitro	Reduced proliferation	----	Nizri et al., 2005
IBU-Octyl-Cytisine	Reduced inflammatory Infiltrates in the spinal cord	Reduced proliferation and proinflammatory cytokine production	----	Niziri et al., 2007
Rivastigmine (AChEI)	Reduced demyelination, microglia activation and axonal damage	Reduced proliferation and proinflammatory cytokine production	----	Nizri et al., 2008
Nicotine	Reduced CNS inflammatory infiltrates, decreased demyelination and reduced axonal loss	Reduced proliferation and proinflammatory cytokine secretion	----	Nizri et al., 2009 Hao et al., 2011
GAT107	Reduced inflammatory infiltrates in the spinal cord	Reduced proliferation and proinflammatory cytokine production	Reduced B- cell number Reduction of B-cell markers’ expression Reduced ab production	Mizrachi et al., 2021

## Data Availability

Upon request from the corresponding authors.

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
