# Peer review of "Neuroinflammation Modulation via α7 Nicotinic Acetylcholine Receptor and Its Chaperone, RIC-3"

_molecules, 2021, doi:10.3390/molecules26206139_

Round 1

Reviewer 1 Report

The manuscript entitled “Neuroinflammation Modulation via a7 Nicotinic Acetylcholine Receptor and Its Chaperone, RIC-3” aims to review numerous effects of α7 nAChR activation on immune cell function and differentiation, and the evidence implicating this receptor and its chaperone RIC-3 in diseases of the central nervous system and neuroinflammation, focusing on MS, AD, and PD. Also, it tries to describe the therapeutic implications of activating the cholinergic anti-inflammatory pathway for diseases involving neuroinflammation. Although the manuscript structure is very good and systematic, not all the parts are covered well. For instance, details on the cell structure and cholinergic system involvement in the imunne system are described suitably. On the other hand, their role in specific diseases (MS, AD, and PD) is not described in enough detail, and it is not clearly indicated. I find sections 8, 9, and 10 too short and not comprehensive enough. Also, the Conclusion part lacks some in-depth analysis and novel, scientifically sound inference. There are only a few references from the last ten years, although the literature is abundant on this topic.

Author Response

1.     "Although the manuscript structure is very good and systematic, not all the parts are covered well. For instance, details on the cell structure and cholinergic system involvement in the immune system are described suitably. On the other hand, their role in specific diseases (MS, AD, and PD) is not described in enough detail, and it is not clearly indicated. I find sections 8, 9, and 10 too short and not comprehensive enough". We did additional literature search, added more details and few new references, in section 8 on multiple sclerosis (page 15-16).Section 9, Parkinson disease, 2 new references were added and more details (page 17-18).Section 10 Alzheimer's disease, page 19. Additional paragraph and 6 new references were added on the study of alpha 7 acetylcholine receptor in AD, in order to present a more extensive view. 2.     "the Conclusion part lacks some in-depth analysis and novel, scientifically sound inference". The conclusions section was extended, (page 20). 3.     "There are only a few references from the last ten years, although the literature is abundant on this topic".  Following the reviewer comment we checked the references list and more than 50% of them are from the last 10 years. 

Reviewer 2 Report

This review is extensive and well organized. The main curtailment is the absence of effects described in humans but otherwise it is a solid manuscript with very minor details which need to be addressed (underlined/commented in appended file).

Once these issues are fixed, it is ready for publication.

Author Response

  1.      Line 35, full name for RIC3 is detailed, for NACHO there is no explanation in the literature for this name
  2.  Line 59, a detail of the cell types was added.
  3.  Line 73, Figure 1 legend was corrected.
  4.  Line 140, SLURP-1 full name provided.
  5.  Line 151-2, detailed.
  6.  Line 301, The title of paragraph 5 was corrected.
  7.  Line 492-4, the legend of table 1 was placed in these lines, after the conclusions, and this is a mistake of the editorial office.
  8.  Line 497, The title of Figure 2 was corrected. 

Reviewer 3 Report

The manuscript titled: “Neuroinflammation Modulation via a7 Nicotinic Acetylcholine Receptor and Its Chaperone, RIC-3” is very interesting and well prepared. The Authors have described in details the role of the receptor for nicotinic acetylcholine receptors and its chaperone RIC-3 in pathology of neurodegenerative disease, including Alzheimer's disease and Parkinson's disease. Additionally, manuscript is enriched in Figures that can help to understand the role of α7 nAChR stimulation in immunomodulation and their effects on selected cells (page 2 and 12). Manuscript is clarify and easy to follow for it.

Author Response

No corrections were required. 

Reviewer 4 Report

Mizrachi et al. review the effects of alpha 7 AChR activation on activation and differentiation of immune cells and their involvement in diseases of the central nervous system.  They also review therapeutic implications of activating this anti-inflammatory pathway.  The review starts with a broad view of AChRs and then focuses on alpha 7 immunomodulatory roles.  The review is focused, concise, and well written.  The complex issues involved are important.

There is a typo in line 434.  In line 136 the number of the reference should be included.

Author Response

The typo mistake in line 434 was corrected
we add the reference number in line 136  

Round 2

Reviewer 1 Report

The authors improved the manuscript according to the comments. I recommend it for publication in the present form.